# Optimizing community linkage to care and antiretroviral therapy Initiation: Lessons from the Nigeria HIV/AIDS Indicator and Impact Survey (NAIIS) and their adaptation in Nigeria ART Surge

Ibrahim Jahun[1]*, Ishaq Said[2], Ibrahim El-Imam[2], Akipu Ehoche[2], Ibrahim Dalhatu[1], Aminu Yakubu[1], Stacie Greby[1], Megan Bronson[3], Kristin Brown[3], Moyosola Bamidele[1], Andrew T. Boyd[3], Pamela Bachanas[3], Emilio Dirlikov[3], Chinedu Agbakwuru[2], Andrew Abutu[1], Michelle Williams-Sherlock[3], Denis Onotu[1], Solomon Odafe[1], Daniel B. Williams[3], Orji Bassey[1], Obinna Ogbanufe[1], Chibuzor Onyenuobi[1], Ayo Adeola[1], Chidozie Meribe[1], Timothy Efuntoye[1], Omodele J. Fagbamigbe[1], Ayodele Fagbemi[1], Uzoma Ene[1], Tingir Nguhemen[1], Ifunanya Mgbakor[1], Matthias Alagi[1], Olugbenga Asaolu[1], Ademola Oladipo[1], Joy Amafah[1], Charles Nzelu[4], Patrick Dakum[5], Charles Mensah[5], Ahmad Aliyu[5], Prosper Okonkwo[6], Bolanle Oyeledun[7], John Oko[8], Akudo Ikpeazu[4], Aliyu Gambo[9], Manhattan Charurat[2], Tedd Ellerbrock[3], Sani Aliyu[9], Mahesh Swaminathan[1]

1 Centers for Disease Control and Prevention-Nigeria Country Office, Abuja, Federal Capital Territory, Nigeria, 2 Maryland Global Initiatives (affiliate of the University of Maryland, Baltimore), Abuja, Federal Capital Territory, Nigeria, 3 Centers for Disease Control and Prevention, Atlanta, Georgia, United States of America, 4 Federal Ministry of Health, Abuja, Nigeria, 5 Institute of Human Virology (IHVN), Abuja, Federal Capital Territory, Nigeria, 6 AIDS Prevention Initiative Nigeria (APIN), Abuja, Federal Capital Territory, Nigeria, 7 Center for Integrated Health Program (CIHP), Abuja, Federal Capital Territory, Nigeria, 8 Catholic Caritas Foundation Nigeria (CCFN), Abuja, Federal Capital Territory, Nigeria, 9 National Agency for the Control of AIDS, Abuja, Federal Capital Territory, Nigeria

* drjahun@yahoo.co.uk

## Abstract

### Background

Ineffective linkage to care (LTC) is a known challenge for community HIV testing. To overcome this challenge, a robust linkage to care strategy was adopted by the 2018 Nigeria HIV/AIDS Indicator and Impact Survey (NAIIS). The NAIIS linkage to care strategy was further adapted to improve Nigeria's programmatic efforts to achieve the 1st 90 as part of the Nigeria Antiretroviral Therapy (ART) Surge initiative, which also included targeted community testing. In this paper we provide an overview of the NAIIS LTC strategy and describe the impact of this strategy on both the NAIIS and the Surge initiatives.

### Methods

The NAIIS collaborated with community-based organizations (CBOs) and deployed mobile health (mHealth) technology with real-time dashboards to manage and optimize community LTC for people living with HIV (PLHIV) diagnosed during the survey. In NAIIS, CBOs' role

**Data Availability Statement:** All relevant data are within the manuscript and its Supporting Information files.

**Funding:** This research has been supported by the President's Emergency Plan for AIDS Relief (PEPFAR) through the Centers for Disease Control and Prevention (CDC) under the terms of CDC Cooperative Agreement No. CDC-RFAGH00-2018-01 and No. CDC-RFAGH17-1753. The findings and conclusions in this report are those of the authors and do not necessarily represent the official position of the funding agencies.

**Competing interests:** The authors have declared that no competing interests exist.

was to facilitate linkage of identified PLHIV in community to facility of their choice. For the ART Surge, we modified the NAIIS LTC strategy by empowering both CBOs and mobile community teams as responsible for not only active LTC but also for community testing, ART initiation, and retention in care.

## Results

Of the 2,739 PLHIV 15 years and above identified in NAIIS, 1,975 (72.1%) were either unaware of their HIV-positive status (N = 1890) or were aware of their HIV-positive status but not receiving treatment (N = 85). Of these, 1,342 (67.9%) were linked to care, of which 952 (70.9%) were initiated on ART. Among 1,890 newly diagnosed PLHIV, 1,278 (67.6%) were linked to care, 33.7% self-linked and 66.3% were linked by CBOs. Among 85 known PLHIV not on treatment, 64 (75.3%) were linked; 32.8% self-linked and 67.2% were linked by a CBO. In the ART Surge, LTC and treatment initiation rates were 98% and 100%, respectively. Three-month retention for monthly treatment initiation cohorts improved from 76% to 90% over 6 months.

## Conclusions

Active LTC strategies by local CBOs and mobile community teams improved LTC and ART initiation in the ART Surge initiative. The use of mHealth technology resulted in timely and accurate documentation of results in NAIIS. By deploying mHealth in addition to active LTC, CBOs and mobile community teams could effectively scale up ART with real-time documentation of client-level outcomes.

## Introduction

Over the last 15 years (2004–2019), Nigeria identified and initiated treatment for at least 1.2 million people living with HIV (PLHIV) [1]. In 2014, the Joint United Nations Program on HIV/AIDS (UNAIDS) set 90-90-90 targets for 2020, whereby 90% of PLHIV are aware of their HIV status (1st 90), 90% of those aware of their HIV status are on antiretroviral therapy (ART) (2nd 90), and 90% of those on ART are virally suppressed (3rd 90) [2]. The 2018 Nigeria HIV/ AIDS Indicator and Impact Survey (NAIIS), a population-based household survey, estimated Nigeria's ARV adjusted progress (by detecting presence of ARVs in blood of all identified PLHIV during NAIIS) towards the UNAIDS 90-90-90 targets to be at 47-96-81 [1]. Nigeria's 1st 90 was among the lowest in Africa [3]. The 1st 90 for West and Central Africa was estimated to be 64% [3]. Before NAIIS, facility-based testing (HIV testing in clinic settings) was used to identify most new PLHIV in Nigeria. After NAIIS, this approach was considered inadequate for accelerating progress toward the 1st 90 targets because of low healthcare facility attendance among Nigerians, especially in rural settings, which have lower socio-economic status when compared to most urban areas [4]. To be successful, any HIV testing approach needed to include an efficient strategy for identifying new PLHIV and linking them to care and treatment [5].

Community-based HIV testing (HIV testing in non-clinic settings) is highly effective in improving uptake of HIV testing services [6]. However, one of the key challenges with this HIV testing approach is linking persons diagnosed with HIV in the community to a health facility for care and treatment. Broadly, these challenges can be divided into three categories,

patient-related barriers, service provision barriers, and documenting and monitoring linkage to care (LTC) barriers. Key patient-related barriers include distance to healthcare facility, transportation costs, stigma and fear of disclosure, denial of HIV diagnosis, and being asymptomatic [7–9]. Key service provision barriers include suboptimal patient-staff relationships, inadequate counseling, absence of clear referral and clinic procedures, disorganized visit schedules, overcrowding, and long waiting times [9]. Documentation and monitoring challenges include use of paper-based records with potential for compromised confidentiality, transcription errors, and duplication of records resulting in inability to determine if PLHIV have successfully linked or to track patients in real-time.

To determine progress toward UNAIDS 90-90-90 target and to characterize the epidemics fully, Nigeria implemented NAIIS in 2018. Following NAIIS, Nigeria realized huge gap in the first 90 target (whereby only 47% are aware of their HIV status) with about 1million PLHIV yet to be identified [1]. To bridge this gap, Nigeria implemented the ART Surge. The NAIIS included community-based HIV testing and implemented a robust linkage to care strategy designed to overcome these barriers related to community-based HIV testing. Similarly, the Nigeria Antiretroviral Therapy (ART) Surge was an ambitious strategy leveraged on NAIIS data to identify and initiate treatment for 500,000 (half of the total first 90 gap) previously undiagnosed PLHIV in 18 months. The ART Surge adopted the linkage to care strategy developed for NAIIS as a complement to community HIV testing activities conducted during this initiative. The strategy primarily focused in six high HIV prevalence states and four states with the highest ART coverage [10]. The aim of this paper is to provide an overview of the structure and achievements of the community linkage to care strategy and describe the impact of this strategy in identifying and linking new PLHIV to care in the NAIIS and the Nigeria Surge initiatives.

## Methods

### Study design and ethical approvals

This was a cross-sectional descriptive study involving PLHIV identified through a population-based household survey—NAIIS (July 2018 –December 2018) across 36 states and the Federal Capital Territory and routine PEPFAR implemented HIV program in Nigeria (October 2019 to April 2020) in nine targeted states. The study was supported by two approved protocols. For the NAIIS, informed consent was obtained, and the protocol was approved by CDC IRB, University of Maryland Baltimore IRB, and Nigeria Health Research Ethics Committee. For procedures and results pertaining to routine PEPFAR program, deidentified data with no perceived ethical risk to participants were collected, so no informed consent was obtained. PEPFAR routine program implementation and ART Surge were reviewed in accordance with CDC human research protection procedures and were determined to be non-research public health program activity and received concurrence from the Government of Nigeria Federal Ministry of Health.

### NAIIS linkage to care (LTC) approach

In NAIIS and ART Surge, a client is linked to care when enrolled and documented at a facility. The NAIIS LTC approach had three components–a) using CBOs to support linkage to care; b) use of mHealth for documentation and monitoring; and c) dashboards for operational management.

**Using CBOs to support linkage to care.** The detailed NAIIS methods have been described elsewhere [1]. All PLHIV identified during NAIIS who self-reported not being on ART at the time of the survey received active or passive linkage to care services. Active linkage

to care (ALTC) involved community-based organizations (CBOs) assisting HIV-positive participants who consented for ALTC to access ART services. The CBOs either provided escort services to the facility or provided them with transport fares and additional support such as psychosocial, education and skills acquisition directly or linking them to such supports if the CBO did not provide these services.

The CBOs were selected based on their experience in HIV program implementation and geographical coverage. All selected CBOs were trained on research ethics, confidentiality and effective HIV counselling skills. One or two CBOs were assigned to each of the 36 states and the Federal Capital Territory depending on the anticipated number of PLHIV to be identified or the geographical size and distribution of the PLHIV. In total, 41 CBOs were selected. In each state, a survey Linkage Coordinator (LC) was assigned to lead and coordinate linkage to care (LTC) activities using a directory of HIV care and treatment facilities. The LC liaised with the CBOs and the person designated as the referral focal staff in each facility to ensure they performed their assigned responsibilities in compliance with NAIIS LTC standard operating procedure. The LC managed all LTC processes using an LC dashboard in real-time. The facility referral focal person managed and completed referrals at facility level by enrolling clients, guiding clients through facility flow for adherence counselling, treatment initiation and other related services. The facility referral focal person also ensured all LTC and treatment initiation processes were documented in real-time.

Passive linkage to care involved the HIV-positive participant accessing ART services without any support (self-linked) but with monitoring.

**Use of mobile health (mHealth) for documentation and monitoring.** mHealth connotes medical or public health service that is facilitated through the use of mobile or wireless devices such as mobile phones, medical sensors, tablets etc [11]. In NAIIS, an mHealth system was deployed to overcome documentation and monitoring challenges linked to community HIV testing and LTC. The mHealth system was managed through the NAIIS Activity Information Management System (AIMS) -a suite of software tools developed to facilitate the administration, collection, and management of the NAIIS survey. With the mHealth system, focal persons at facilities to which persons diagnosed with HIV in the field are referred can send confirmation of successful referral using USSD (Unstructured Supplementary Service Data). There is also a full stack web-based application with a relational database that allows system administrators (including LCs and CBOs) to assign participants to CBO personnel and for CBOs to document and monitor LTC processes via their respective dashboards. The USSD-based part was integrated with all existing mobile network providers and shared the same database as the web component. The whole module was integrated, in a unidirectional manner, with the AIMS database for information exchange. The mHealth database and transmission of data involving it were protected using industry standard security protocols including firewalls, secure file transfer protocols and encryption as applicable. Only authorized survey personnel were provided access to the database.

## NAIIS ALTC operational management structure, tools and dashboards

Linkage to care processes started with the counsellor-interviewer (Fig 1) who provided counselling to participants before and after testing. The counsellor-interviewer was responsible for a) referring HIV-positive participants to healthcare facilities of their choice from a directory of HIV care and treatment facilities, using approved paper-based referral forms and b) completion of the electronic consent and participant locator forms for each participant who consented for ALTC. Only participants who consented to sharing personal information with CBOs were provided ALTC. Information entered into the electronic participant locator form appears on

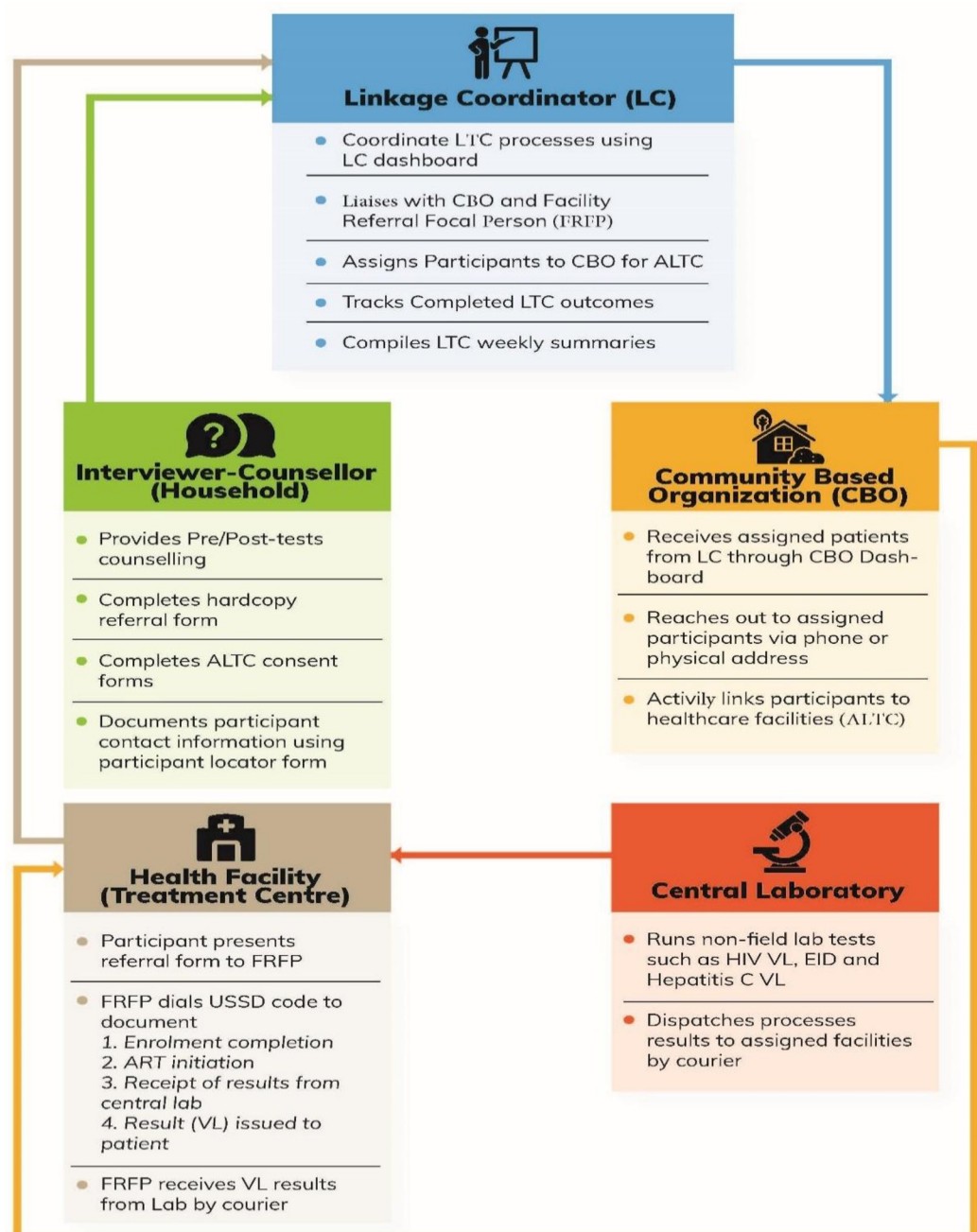

**Fig 1. Active linkage to care processes, NAIIS 2018.** Abbreviations: LC–Linkage Coordinator; LTC–Linkage to Care; CBO–Community Based Organization; ALTC–Active Linkage to Care; FRFP–Facility Referral Focal Person; USSD–Unstructured Supplementary Service Data; VL–Viral Load; EID–Early Infant Diagnosis.

the LC dashboard in near real-time and is used by the CBO for active tracking of participants who consented for ALTC. Participants were allowed seven days to visit a health care facility through passive linkage. Those who did not visit a health care facility within seven days were assigned to a CBO by the LC using the LC dashboard to start active linkage to care.

Once assigned to a CBO, participant information on the LC dashboard was available on the CBO dashboard. Upon receipt of the information, the CBO contacted the participant,

developed a linkage plan, and initiated ALTC processes. When at their chosen health facility, regardless of mode of linkage (active or passive), participants presented the facility referral focal person the hardcopy referral form they received from the counselor-interviewer with the participant's unique identification number (unique ID) and their HIV-positive test result. There is no difference in documentation and enrollment of participants referred through the two LTC approaches at the facility by the facility referral focal person. Participants referred through ALTC will appear in both CBO and LC dashboards while those referred through passive linkage will only appear in the LC dashboard. For all participants, an HIV rapid test using the same testing algorithm as that in the field was repeated to confirm the positive test result and a plan of action was agreed upon between the participant and the clinician. Following enrollment at the facility, the participant is considered as a registered (enrolled) patient and in line with National ART Guidelines, the participant is initiated on ART and scheduled for routine periodic ARV refills and monitoring at the facility. LTC outcomes (option 1 –participant enrolled, option 2 –participant commenced ART) were documented by the facility referral focal person using a handset and a combination of USSD code and participant's unique ID.

## Surge adaptation of NAIIS linkage to care strategy

The Surge started in April 2019 and expected to continue until at least 500,000 new PLHIV are identified and initiated on ART. Community testing started October 2019. To improve effectiveness of linking clients that test positive to care in the community, two models of linkage were adopted; 1) partnering with CBOs as was done in NAIIS and 2) use of mobile community testing and linkage to care teams, a new and complementary strategy. This latter strategy involved constituting teams of testers and counselors (who also served as linkage to care focal persons) offering rapid HIV antibody testing in line with national testing algorithm in the community and actively linking HIV-positive individuals to healthcare facilities. A team is made up of two counselors and a tester. Both models relied on ALTC and were implemented in line with the lessons learnt from NAIIS–client escort, provision of transport fares to those who needed, client choice of treatment facility from a list of available healthcare facilities, and provision of additional support such as psychosocial, education and skills acquisition support or linking them to such supports if the CBOs do not provide such additional services. In the Surge, participants were not offered the option of passive referral because the Surge was not a study–it was an intervention aimed at fast tracking Nigeria's progress towards achieving the 1st 90 and lessons from the NAIIS LTC had shown ALTC to be superior to passive linkage.

In the Surge, CBOs conducted ALTC and provided a package of community services including case identification, linkage with treatment initiation, and retention. Clients in hard-to-reach locations could initiate ART in the community either through CBOs or through mobile community testing and LTC teams. Community treatment initiation involved provision of 3-month ARV starter pack (first ART dispense) right in the community to newly identified PLHIV who presented with no symptoms and reported being stable in line with the National Task Shifting and Task Sharing Policy. Upon utilization of the starter pack, the client has the option of getting refills in the community or visiting the facility of his/her choice from the directory of treatment facilities if there is a medical concern. The CBOs and the mobile community teams were responsible for their assigned clients throughout the continuum of care, including ensuring the clients were retained. Retention was defined as patient's continued engagement in care and can be measured for a specific period: for example, 3 months (three-month retention). The CBOs and mobile community teams are qualified non-physician healthcare workers that received special training on LTC and community treatment initiation from the central Surge team prior to deployment.

The mHealth strategy was not used during the Surge because there was insufficient time to deploy the system prior to Surge initiation. Rather, data was collected manually using national HIV testing data collection tools which included completion of a "client-intake form" in the field for each HIV-positive individual and documentation of enrollment into the national ART register at the healthcare facility. Daily data reconciliation was conducted by the Monitoring and Evaluation (M&E) Officer and field testers to compile weekly LTC summaries.

### Linkage to care indicators and data analysis for NAIIS and Surge

In NAIIS, the number of individuals identified with HIV, including both newly diagnosed and known PLHIV, were monitored daily. The analysis in this paper is based on the de facto population of survey participants. Among the known PLHIV, those who reported being on ART were not included in the analyses since they were not provided linkage to care services. Additionally, the number of clients linked to care (ALTC or passive linkage), and number of clients that were initiated on ART were monitored. Variability in LTC and ART initiation rates between active versus passive LTC approaches among known PLHIV and newly identified PLHIV populations were stratified by sex and compared using McNemar's test of proportion.

In the Surge, the number of new PLHIV identified, number of PLHIV linked to care and number of PLHIV initiated on ART were monitored weekly. Linkage to care rate was measured as proportion of newly identified PLHIV who were enrolled in to care (numerator) to the total number of new PLHIV identified (denominator). ART initiation rate was measured as proportion of newly identified PLHIV who were linked to care and initiated ART (numerator) to the number of PLHIV enrolled into care (denominator). To ensure those individuals who were provided ART at the time of diagnosis in facilities and in community settings truly started ART, three-month retention was calculated for each monthly cohort of newly identified PLHIV to evaluate the overall impact of the modified NAIIS linkage to care approach. The proportion of the number of individuals still on treatment at 3 months (numerator) after initiating ART to the number of individuals initiated on ART at month 0 (denominator) was determined as three-month retention and was measured when the client returned for second ARV refill. Linkage to care data for the first 30 weeks of community program implementation (October 2019 –May 2020) were included in the analysis.

## Results

Of 2,739 PLHIV age 15 years and above identified during NAIIS, 1,890 (69%) were newly diagnosed PLHIV and 849 (31%) were known PLHIV who were re-confirmed as HIV positive during the survey, of which 85 (10%) reported not being on ART. Of the newly diagnosed PLHIV, 1,278 (67.6%) were linked to care (Table 1), with more using ALTC through CBOs (66.3%) than using passive linkage (33.7%). Similarly, more PLHIV were linked to care using ALTC than passive linkage among known PLHIV not on ART (67.2% and 32.8%), respectively. Of the total PLHIV linked to care, 952 (70.9%) were initiated on ART. ART initiation rates among newly identified PLHIV and known PLHIV not on ART were 70.7% and 75.0%, respectively (Table 1).

During the first 30 weeks of the ART Surge, a total of 71,995 PLHIV were identified with 39,288 (54.6%) PLHIV identified through community testing. Weekly case identification ranged from 156 in week 1 to 2,277 in week 30. There was an observed increase in the contribution from community case finding from 8.0% in week 1 to 80.2% in week 30. Of the total 39,288 PLHIV identified in the community, 38,504 were linked to care, resulting in a 98% linkage rate, range (week 10 [94%]–week 17 [100%]). We observed 100% weekly ART initiation rate in each of the 30 weeks of the surge (Fig 2). Of the 38,504 who initiated ART, only 3,404 (8.8%) were from hard-to-reach locations and hence eligible for community treatment

**Table 1. People living with HIV (PLHIV) age ≥15 years linkage to care (LTC) and antiretroviral therapy (ART) initiation by category of self-report HIV status and sex, NAIIS 2018.**

| | HIV infection identified | Linkage to care | | | ART initiation | | |
|---|---|---|---|---|---|---|---|
| | N | N | Linkage rate (%) | p† | N | ART initiation rate (%) | p† |
| Overall | 1,975 | 1,342 | 67.9 | | 952 | 70.9 | |
| **Sex** | | | | 0.357 | | | 0.500 |
| Male | 611 | 424 | 69.4 | | 306 | 72.2 | |
| Female | 1,364 | 918 | 67.3 | | 646 | 70.4 | |
| **Case type** | | | | 0.138 | | | 0.463 |
| Newly diagnosed | 1,890 | 1,278 | 67.6 | | 904 | 70.7 | |
| Known infection not on ART | 85 | 64 | 75.3 | | 48 | 75.0 | |

† p-value for the chi-square test statistic.

initiation. Three-month retention for the monthly treatment initiation cohorts showed progressive improvement in retention: 76% (October 2019 treatment cohort, 3-month retention evaluation was in January 2020) to 90% (April 2020 treatment cohorts, 3-month retention evaluation was in July 2020) (Fig 2).

## Discussions

Actively monitoring and evaluating linkage to care data during NAIIS and the Surge were associated with a high percentage of PLHIV accessing ART through active LTC interventions.

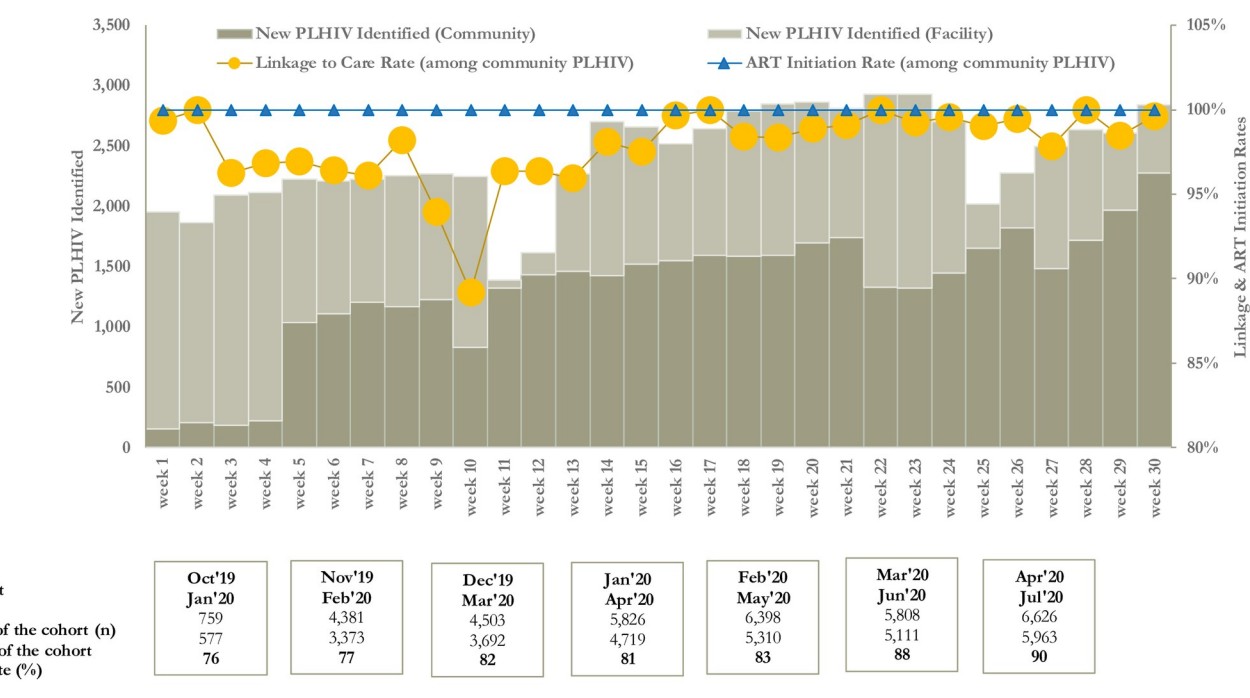

**Fig 2. Linkage, Antiretroviral (ART) initiation, and retention rates during ART Surge, Nigeria, October 2019 –May 2020.** Abbreviations: PLHIV–People Living with HIV; ART–Antiretroviral Therapy; HTS–HIV Testing Service.

Using the mHealth strategy in NAIIS during the short time the large survey was in the field, we were able to efficiently monitor and manage all PLHIV identified from time of HIV diagnosis to ART initiation, using CBOs to ensure those not accessing services within one week were provided assistance. Leveraging this same network of CBOs during the ART Surge with close tracking of all identified PLHIV to ensure enrollment in care and initiation on treatment and client-centered services to those who needed assistance may have resulted in the high linkage to care rate of 98%.

The linkage and ART initiation rates achieved by the NAIIS model were much higher than than LTC rates obtained in conventional HIV community outreaches in Nigeria [12]. For example, a study in Rivers State indicated linkage to care rates of 25% and 43% for passive and community ART respectively [12]. Similarly, in rural Tanzania, community linkage to care was as low as 28% due to stigma, denial and gap in knowledge and awareness of the value of early linkage to care [13]. In a provincial surveillance survey in Kwazulu-Natal, South Africa, the community linkage to care rate was estimated at 55% with 21% of eligible participants initiating ART [14]. The higher linkage rates reported among those who were actively linked can be attributed to the reported effectiveness of using CBOs in community health initiatives, which help with stigma reduction that serves as major obstacle for successful linkage to care [15,16]. In both the NAIIS and the ART Surge, the CBOs that supported active linkage to care provided interim solutions to major linkage to care barriers. Some of these solutions included support for transport and/or escort to health facilities to facilitate linkage, as well as provision of other supportive services, such as psychosocial and education support. This is consistent with other studies that have found these interventions to be effective for improving LTC [17–19]. It is likely that involving CBOs in linkage to care also reduced the level of mistrust and reservations observed in communities, whenever community health activities including unfamiliar persons, such as NAIIS staff, enter communities.

Among NAIIS participants testing HIV-positive, the difference in linkage rates between those who reported knowing their HIV-positive status and those who did not can be explained in various ways. The higher linkage to care rates observed among previously known HIV-positive participants not on ART may be attributed to the reconfirmation of positive test results during the survey because they didn't think they were positive before. It may also be that the CBOs providing active support and interventions helped these persons to overcome barriers to access that may have hindered their linkage to care in the past. On the other hand, the lower linkage rates seen among those who did not know their status may have been either because they were known HIV-positive persons on ART but refused to disclose their HIV-positive status during the survey, asymptomatic, or not ready to accept their HIV-positive status. A study from South Africa reported that PLHIV who reported being in good health were less likely to link to care [20]. Similarly, denial was shown to be an important factor impeding linkage to care and treatment initiation; acceptance of a positive result and disclosure are associated with higher rate of linkage to care [21,22].

Following scale-up of community testing through engagement of more CBOs and community testing teams, we observed progressive increases in the community contribution to overall case finding efforts of almost 10 fold from week 1 to week 30 (8.0% versus 80.2%). Targeted community testing was shown to be an efficient way of rapid identification of new PLHIV in Nigeria [23]. The high linkage and treatment initiation rates observed in the Nigeria ART Surge might have been influenced by four key changes implemented during the Surge based on lessons learned from NAIIS linkage strategies: 1) active strategies to link PlHIV to care and treatment services were used 2) mobile community testing and linkage teams were deployed to complement the CBOs' efforts. as the Surge covers larger geographical locations, which available CBOs were not able cover alone, 3) CBOs and community testing teams were responsible

for ensuring all PLHIV identified were initiated and retained on treatment, and 4) PLHIV in hard-to-reach areas had the opportunity to initiate ART in the community. To ensure that PLHIV who were identified, linked and initiated ART were retained, patient education and adherence counselling were emphasized during training of CBOs and community teams and implementation was monitored during the Surge. Therefore, it is likely that CBOs provided useful counselling and patient education on the benefits of early and consistent treatment, which might have influenced PLHIV to accept linkage and treatment initiation since good client education, awareness, and overall understanding of treatment benefits and needs for life-long treatment have been shown to improve treatment initiation uptake and retention [24,25].

Although 100% of new and known HIV-positive persons identified in the Surge who were not on ART were offered treatment, the 3-month retention data suggest that not all of them started or remained on treatment. These data suggest that additional early follow-up of newly initiated persons may be necessary to ensure that they start and remain on treatment. We observed progressive improvement in three-month retention among monthly cohorts of PLHIV who were linked and initiated on ART in the Surge. During the Surge, CBOs and community teams deployed innovative strategies to ensure optimum retention. An important treatment attrition preventive strategy we implemented was keeping in touch with patients from time of treatment initiation until the next refill period by frequent calls or SMS messages to provide adherence couseling, patient education, and support. Early retention data after 3 months suggests that some patients may not have initiated treatment even though they took the drugs or may have started and then stopped in a short period of time and therefore, more research is needed to determine the best interventions for newly identified PLHIV to support durable linkage and early retention as this group has been shown to be 2–3 times more likely to experience an interruption in treatment than those who have been on ART >3 months [26]. Studies have shown that bringing treatment closer to PLHIV substantially improves linkage and treatment initiation among linkage-resistant PLHIV [27–29], therefore the community treatment initiation strategy deployed to hard-to-reach areas might have also contributed to higher linkage rates in the Surge, even though the proportion of PLHIV who were initiated on treatment in the community was low.

The use of USSD code and dashboards by LC and CBO as part of the NAIIS LTC strategy (not used in ART Surge) served as a solution to the third category of barriers to linkage to care (problems documenting linkage to care). Unfortunately, this strategy was not used in the Surge due to time and funding constraints. Manual linkage to care management using registers and daily meetings for records reconcilliation in the Surge was found to be cumbersome and inefficient. Daily and weekly meetings to reconcile LTC records between CBOs, community teams and sites takes several man-hours which could have been used to find and link more PLHIV to lifesaving ART. Similarly, dealing with hardcopy records put data security and confidentiality at higher risk when compared with the mHealth strategy. Data quality challenges, including transcription errors, are common such as age and sex miscategorization and various inconsistencies which may affect quality of evaluations. Additionally, these errors result in challenges tracking and tracing individuals for linkage and retention efforts and determining which patients are lost in the system. Following the lessons learned from NAIIS in using the mHealth technology for LTC documentation, and the inefficiencies experienced in manual LTC documentation in the Surge, there are plans to deploy mHealth technology for the Surge going forward to enhance efficiencies in Surge LTC and field operations. Using mHealth, NAIIS staff ensured that unique individuals were counted at sites after referral and patients linked passively or by CBOs did not register in more than one healthcare facility. Several other benefits of using mHealth have been reported in efforts at improving efficiencies in Early Infant Diagnosis (EID) turnaroud time in some developing countries [30], improving

accuracy, completeness, and timeliness of reporting from healthcare facilities to district health ministry data repositories [31].

The generalization of our experience in optimizing LTC through active linkage with CBOs was subject to at least three limitations. First, this strategy was designed as a program and not as an experiment. Thus, the strategy was not designed to evaluate the superiority of our package in comparison to existing practice. The absence of a rigorous analysis to control for potential socio-economic and socio-behavioural confounders that would have been in place, if this was designed as an experiment, may affect the replicability of our experience elsewhere. Second, which is closely related to the first limitation, is that we were unable to ascertain the impact of individual strategies deployed by the CBOs on linkage to care and treatment initiation rates and outcomes. Third, categorization into known PLHIV and newly identified PLHIV was based on participant self-report and may be misleading. Several studies, including NAIIS, have reported low disclosure rates among PLHIV during surveys or in healthcare settings [1,32–34]. Therefore it may be possible that some of the newly identified PLHIV might actually be known PLHIV and on treatment hence refused linkage to care.

## Conclusions

Active linkage to care using CBOs was associated with high levels of community linkage to care and ART initiation among PLHIVs in the NAIIS and the Nigeria ART Surge. In addition, the use of mHealth technology in the NAIIS linkage to care strategy was associated with timely documentation of results with high accuracy. By deploying active LTC strategies with mHealth technology, CBOs and mobile community teams may be able to effectively and efficiently scale-up ART with real-time documentation of client level outcomes across disparate service delivery points in low-resource settings like Nigeria.

## Supporting information

**S1 File.**
(XLSX)

## Acknowledgments

The authors acknowledge the enormous contributions from the NAIIS consortium (Center for International Health, Education and Biosecurity [CIHEB] at the University of Maryland, African Field Epidemiology Network [AFENET], University of Washington [UW] and ICF International), the nine state ministries of health (Benue, Delta, Enugu, FCT, Gombe, Imo, Lagos, Nasarawa, Rivers) and the four PEPFAR implementing partners (APIN Public Health Initiative [APIN], Center for Integrated Health Program in Nigeria [CIHP], Catholic Caritas Foundation Nigeria [CCFN], Institute of Human Virology Nigeria [IHVN]).

## Author Contributions

**Conceptualization:** Ibrahim Jahun, Ishaq Said, Ibrahim El-Imam, Ibrahim Dalhatu, Aminu Yakubu, Stacie Greby, Megan Bronson, Kristin Brown, Pamela Bachanas.

**Data curation:** Ibrahim Jahun, Akipu Ehoche.

**Formal analysis:** Ibrahim Jahun, Ishaq Said, Ibrahim El-Imam, Akipu Ehoche, Moyosola Bamidele.

**Investigation:** Ibrahim Jahun.

**Methodology:** Ibrahim Jahun, Ibrahim El-Imam, Aminu Yakubu, Megan Bronson, Andrew T. Boyd, Emilio Dirlikov, Chinedu Agbakwuru, Solomon Odafe.

**Software:** Akipu Ehoche, Moyosola Bamidele.

**Supervision:** Ibrahim Jahun, Ibrahim Dalhatu, Stacie Greby, Megan Bronson, Kristin Brown, Emilio Dirlikov, Tedd Ellerbrock, Mahesh Swaminathan.

**Validation:** Ibrahim Jahun, Ishaq Said, Ibrahim El-Imam, Aminu Yakubu, Megan Bronson, Andrew T. Boyd, Pamela Bachanas, Andrew Abutu, Michelle Williams-Sherlock, Denis Onotu, Solomon Odafe, Daniel B. Williams, Orji Bassey, Obinna Ogbanufe, Chibuzor Onyenuobi, Ayo Adeola, Chidozie Meribe, Timothy Efuntoye, Omodele J. Fagbamigbe, Ayodele Fagbemi, Uzoma Ene, Tingir Nguhemen, Ifunanya Mgbakor, Matthias Alagi, Olugbenga Asaolu, Ademola Oladipo, Joy Amafah, Charles Nzelu, Charles Mensah, Ahmad Aliyu, Prosper Okonkwo, Bolanle Oyeledun, John Oko, Akudo Ikpeazu, Aliyu Gambo, Manhattan Charurat, Tedd Ellerbrock, Sani Aliyu, Mahesh Swaminathan.

**Visualization:** Ibrahim Jahun, Chinedu Agbakwuru, Patrick Dakum.

**Writing – original draft:** Ibrahim Jahun.

**Writing – review & editing:** Ibrahim Jahun, Ishaq Said, Ibrahim El-Imam, Akipu Ehoche, Ibrahim Dalhatu, Aminu Yakubu, Stacie Greby, Megan Bronson, Kristin Brown, Moyosola Bamidele, Andrew T. Boyd, Pamela Bachanas, Emilio Dirlikov, Chinedu Agbakwuru, Andrew Abutu, Michelle Williams-Sherlock, Denis Onotu, Solomon Odafe, Daniel B. Williams, Orji Bassey, Obinna Ogbanufe, Chibuzor Onyenuobi, Ayo Adeola, Chidozie Meribe, Timothy Efuntoye, Omodele J. Fagbamigbe, Ayodele Fagbemi, Uzoma Ene, Tingir Nguhemen, Ifunanya Mgbakor, Matthias Alagi, Olugbenga Asaolu, Ademola Oladipo, Joy Amafah, Charles Nzelu, Patrick Dakum, Charles Mensah, Ahmad Aliyu, Prosper Okonkwo, Bolanle Oyeledun, John Oko, Akudo Ikpeazu, Aliyu Gambo, Manhattan Charurat, Tedd Ellerbrock, Sani Aliyu, Mahesh Swaminathan.

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
