## [Decision Letter · Decision Letter 0]

6 Aug 2021

PONE-D-21-15583

Optimizing community linkage to care and antiretroviral therapy Initiation: Lessons from the Nigeria HIV/AIDS Indicator and Impact Survey (NAIIS) and their adaptation in Nigeria ART Surge.

PLOS ONE

Dear Dr. Ibrahim Jahun,

Thank you for submitting your manuscript to PLOS ONE. After careful consideration, we feel that it has merit but does not fully meet PLOS ONE’s publication criteria as it currently stands. Therefore, we invite you to submit a revised version of the manuscript that addresses the points raised during the review process.

We look forward to receiving your revised manuscript.

Kind regards,

Professor Kwasi Torpey, MD PhD MPH

Academic Editor

PLOS ONE

“This research has been supported by the President’s Emergency Plan for AIDS Relief (PEPFAR) through the Centers for Disease Control and Prevention (CDC) under the terms of CDC Cooperative Agreement No. CDC-RFAGH17-1753.“

“This research has been supported by the President’s Emergency Plan for AIDS Relief (PEPFAR) through the Centers for Disease Control and Prevention (CDC) under the terms of CDC Cooperative Agreement No. CDC-RFAGH00-2018-01 and No. CDC-RFAGH17-1753. The findings and conclusions in this report are those of the authors and do not necessarily represent the official position of the funding agencies.”

4. We note that you have referenced : Dirlikov E, Jahun I, Odafe S. F, et al. (2021). Rapid Scale-up of an Antiretroviral Therapy Program Before and During the COVID-19 Pandemic — Nine States, Nigeria, March 31, 2019–September 30, 2020 (MMWR – unpublished) and Jahun I, Dirlikov E, Odafe S. F et al. (2021). Ensuring Efficient Targeted Community Testing Services in Low HIV Prevalence Settings in Nigeria, October 2019–March 2020: Acceleration to HIV Epidemic Control (unpublished paper).

which has currently not yet been accepted for publication. Please remove this from your References and amend this to state in the body of your manuscript: (ie “Bewick et al. [Unpublished]”) as detailed online in our guide for authors

http://journals.plos.org/plosone/s/submission-guidelines#loc-reference-style.

Additional Editor Comments (if provided):

Reviewers' comments:

Reviewer's Responses to Questions

**Comments to the Author**

1. Is the manuscript technically sound, and do the data support the conclusions?

Reviewer #1: Yes

Reviewer #2: Yes

2. Has the statistical analysis been performed appropriately and rigorously? 

Reviewer #1: Yes

Reviewer #2: Yes

3. Have the authors made all data underlying the findings in their manuscript fully available?

Reviewer #1: Yes

Reviewer #2: Yes

4. Is the manuscript presented in an intelligible fashion and written in standard English?

Reviewer #1: Yes

Reviewer #2: Yes

5. Review Comments to the Author

Reviewer #1: This article is an important contribution to improving the continuum of care from HIV testing to treatment. This article reports on two programs and sources of data: NIIS and Surge ART, and their respective efforts to increase community-based HIV testing, linkage to care, and ART initiation.

General comments: 1) the article provided a lot of information to digest for readers unfamiliar to these programs, and the article can benefit from some structuring to introduce and link these programs; 2) the article can be written more concisely and with some copy/editing. Where possible, reduce on the use of acronyms as it makes it very difficult to read.

Detailed comments:

- Line 52: Correct years (2024)

- Line 78-79: Articulate how documentation and monitoring factors affected LTC exactly

- Line 106: Clarify difference between linkage coordinator and the counsellor-interviewer? And what is the role of the FRFP if the CBO is doing this?

- Line 271: Noted change from passive to active LTC - in NIIS you gave people a choice, in Surge ART you weren't require to give people a choice because it was not a study setting?

- Line 271: How did mobile community testing and linkage team's role complement CBO's efforts? Tease out more specifically? (If you were to do this in another setting, would you still have CBO? Did these 2 groups play unique roles?)

- CTI strategy - starter pack ART is for how long? How is client linked to his/her static clinic? Or do you notice attrition?

- Can you discuss the policy environment that allows for CTI? Anything unique about composition of mobile team/ CBOs that allows for roll out of CTI?

Thank you.

Reviewer #2: 1. The background information as shown in the abstract (Lines 29-33) suggest that the focus of the work is on ART Surge. However, the paper seems more focused on NAIS and not ART Surge. In addition, the authors are using the results of the 2018 NAIS and a basis to provide an overview of the NAIIS LTC strategy and describe the impact in both the NAIIS (2018) and the Surge. I think the background need some finetuning to give some clarity.

2. In addition to the 3-month Retention rate, it would have been good if the authors could also determine 6-month Retention rate for the cohorts that are eligible (those that initiated treatment/ART from Oct-19 to Jan-20) this will help determine if the PLHIV are being retained in care for a longer time.

3. The reason stated in lines 252-254: “On the other hand, the lower linkage rates seen among those who did not know their status may have been either because they were known HIV-positive participants on ART but refused to disclose their HIV-positive status during the survey, asymptomatic, or not ready to accept their HIV-positive status” seems to be inconsistent with the statement in lines 55-58 to the effect that the presence of ARVs in blood of all identified PLHIV during NAIIS was undertaken). If NAIS checked for the presence of ARV in the blood of all identified PLHIV, then it will be possible to identify those who were already positive and refused to disclose their HIV-positive status during the survey through the unique ID.

4. Typographical errors

Line 52 (2024-2019) should be corrected

Reference in line 241 (154) should be corrected.

6. PLOS authors have the option to publish the peer review history of their article (what does this mean?). If published, this will include your full peer review and any attached files.

Reviewer #1: No

Reviewer #2: No

---

## [Author Response · Author response to Decision Letter 0]

29 Aug 2021

Response to Academic Editor’s comments/queries:

Query 1:

Response: Thank you for your observations. The authors have downloaded PLOS ONE style template and format and ensured that the revised paper is well aligned with the requirements.

Query 2:

We note that the grant information you provided in the ‘Funding Information’ and ‘Financial Disclosure’ sections do not match. When you resubmit, please ensure that you provide the correct grant numbers for the awards you received for your study in the ‘Funding Information’ section.

Response: Thank you for your observations and apologies for the mismatch which was due to typographic error. This is now corrected, and the correct grant information is: “This research has been supported by the President’s Emergency Plan for AIDS Relief (PEPFAR) through the Centres for Disease Control and Prevention (CDC) under the terms of CDC Cooperative Agreement No. CDC-RFAGH00-2018-01 and No. CDC-RFAGH17-1753”

Query 3:

Thank you for stating the following in the Acknowledgments Section of your manuscript:

“This research has been supported by the President’s Emergency Plan for AIDS Relief (PEPFAR) through the Centres for Disease Control and Prevention (CDC) under the terms of CDC Cooperative Agreement No. CDC-RFAGH17-1753. “

We note that you have provided funding information that is not currently declared in your Funding Statement. However, funding information should not appear in the Acknowledgments section or other areas of your manuscript. We will only publish funding information present in the Funding Statement section of the online submission form. Please remove any funding-related text from the manuscript and let us know how you would like to update your Funding Statement. Currently, your Funding Statement reads as follows:

“This research has been supported by the President’s Emergency Plan for AIDS Relief (PEPFAR) through the Centres for Disease Control and Prevention (CDC) under the terms of CDC Cooperative Agreement No. CDC-RFAGH00-2018-01 and No. CDC-RFAGH17-1753. The findings and conclusions in this report are those of the authors and do not necessarily represent the official position of the funding agencies.”

Response: Thank you for your observations.

• Funding information is removed from the manuscript.

• The correct funding statement should be: “This research has been supported by the President’s Emergency Plan for AIDS Relief (PEPFAR) through the Centres for Disease Control and Prevention (CDC) under the terms of CDC Cooperative Agreement No. CDC-RFAGH00-2018-01 and No. CDC-RFAGH17-1753. The findings and conclusions in this report are those of the authors and do not necessarily represent the official position of the funding agencies.”

Query 4:

We note that you have referenced: Dirlikov E, Jahun I, Odafe S. F, et al. (2021). Rapid Scale-up of an Antiretroviral Therapy Program Before and During the COVID-19 Pandemic — Nine States, Nigeria, March 31, 2019–September 30, 2020 (MMWR – unpublished) and Jahun I, Dirlikov E, Odafe S. F et al. (2021). Ensuring Efficient Targeted Community Testing Services in Low HIV Prevalence Settings in Nigeria, October 2019–March 2020: Acceleration to HIV Epidemic Control (unpublished paper), which has currently not yet been accepted for publication. Please remove this from your References and amend this to state in the body of your manuscript: (ie “Bewick et al. [Unpublished]”) as detailed online in our guide for authors.

Response: Thank you for the guidance. Kindly note:

• Dirlikov E, Jahun I, Odafe S. F, et al. (2021). Rapid Scale-up of an Antiretroviral Therapy Program Before and During the COVID-19 Pandemic — Nine States, Nigeria, March 31, 2019–September 30, 2020 is now published and reference is updated accordingly. https://www.cdc.gov/mmwr/volumes/70/wr/mm7012a3.htm

• Jahun I, Dirlikov E, Odafe S. F et al. (2021). Ensuring Efficient Targeted Community Testing Services in Low HIV Prevalence Settings in Nigeria, October 2019–March 2020: Acceleration to HIV Epidemic Control. This paper is also now published and reference is updated accordingly. https://www.dovepress.com/articles.php?article_id=68206.

Response to Reviewer 1 comments/queries:

General comments: 

1. the article provided a lot of information to digest for readers unfamiliar to these programs, and the article can benefit from some structuring to introduce and link these programs. 

Response: Thank you for your observations and recommendations. The article is edited and restricted and the two programs are now well integrated. The flow has now improved substantially.

2. the article can be written more concisely and with some copy/editing. Where possible, reduce on the use of acronyms as it makes it very difficult to read.

Response: Thank you for your observations and recommendations. The authors have reworked the article in response to your recommendations. 

Detailed comments:

• Line 52: Correct years (2024)

Response: Thank you for the observation. years corrected (2004 – 2019), see line 57.

• Line 78-79: Articulate how documentation and monitoring factors affected LTC exactly

Response: Thank you for the observation. Here is the updated versions: “Documentation and monitoring challenges include use of paper-based records with potential for include compromised confidentiality, transcription errors, and duplication of records resulting in inability to determine if PLHIV have successfully linked or linkage success and to track patients in real-time.”. See line 84 – 87

• Line 106: Clarify difference between linkage coordinator and the counsellor-interviewer? And what is the role of the FRFP if the CBO is doing this?

Response: Thank you for the comment. Kindly see edits from line 104 – 108. “In each state, a survey Linkage Coordinator (LC) was assigned to lead and coordinate linkage to care (LTC) activities using a directory of HIV care and treatment facilities. The LC liaised with the CBOs and the person designated as the referral focal staff in each facility to ensure they performed their assigned responsibilities in compliance with NAIIS LTC standard operating procedure. The LC managed all LTC processes using an LC dashboard in real-time. The facility referral focal person managed and completed referral at facility level by enrolling client, guiding client through facility flow for adherence counselling, treatment initiation and other related services. The facility referral focal person also ensured all LTC and treatment initiation processes are well documented in real-time”. See line 138 – 148

• Line 271: Noted change from passive to active LTC - in NIIS you gave people a choice, in Surge ART you weren't require to give people a choice because it was not a study setting?

Response: Thank you for your question. Kindly see edits from line 224 – 227. “In the Surge, participants were not offered the option of passive referral because the Surge was not a study – it was an intervention aimed at fast tracking Nigeria’s progress towards achieving the 1st 90 and lessons from the NAIIS LTC had shown ALTC to be superior to PLTC”.

• Line 271: How did mobile community testing and linkage team's role complement CBO's efforts? Tease out more specifically? (If you were to do this in another setting, would you still have CBO? Did these 2 groups play unique roles?)

Response: Thanks for your question. Kindly see edits in line 172. Please see line 350 – 352. “This was because unlike NAIIS, which is limited to selected enumeration areas, the Surge covers larger geographical locations, which available CBOs were not able cover alone”

• CTI strategy - starter pack ART is for how long? How is client linked to his/her static clinic? Or do you notice attrition?

• Can you discuss the policy environment that allows for CTI? Anything unique about composition of mobile team/ CBOs that allows for roll out of CTI?

Response (for both CTI strategy and CTI policy): Thank you for your questions. Kindly see edits from line 231 -237. “CTI involved the provision of 3-month ARV starter pack (first ART dispense) right in the community to newly identified PLHIV who presented with no symptoms and reported being stable, in line with the National Task Shifting and Task Sharing Policy. Upon utilization of the starter pack, the client has the option of getting refill in the community or to visit the facility of his/her choice from the directory of treatment facilities if there is a medical concern. The CBOs and the mobile community teams were responsible for their assigned clients throughout the continuum of care, including ensuring the clients were retained.”

Response to Reviewer 2 comments/queries:

• The background information as shown in the abstract (Lines 29-33) suggest that the focus of the work is on ART Surge. However, the paper seems more focused on NAIS and not ART Surge. In addition, the authors are using the results of the 2018 NAIS and a basis to provide an overview of the NAIIS LTC strategy and describe the impact in both the NAIIS (2018) and the Surge. I think the background need some finetuning to give some clarity.

Response: Thank you for the very important feedback. This is well noted and addressed in the abstract and the background is substantially edited to address your observations.

• In addition to the 3-month Retention rate, it would have been good if the authors could also determine 6-month Retention rate for the cohorts that are eligible (those that initiated treatment/ART from Oct-19 to Jan-20) this will help determine if the PLHIV are being retained in care for a longer time.

Response: Thank you for your recommendation. This aspect is being addressed in another paper which investigated retention in Surge. The paper has been approved by CDC science office and is submitted to a journal. The paper will refence this paper once accepted.

• The reason stated in lines 252-254: “On the other hand, the lower linkage rates seen among those who did not know their status may have been either because they were known HIV-positive participants on ART but refused to disclose their HIV-positive status during the survey, asymptomatic, or not ready to accept their HIV-positive status” seems to be inconsistent with the statement in lines 55-58 to the effect that the presence of ARVs in blood of all identified PLHIV during NAIIS was undertaken). If NAIS checked for the presence of ARV in the blood of all identified PLHIV, then it will be possible to identify those who were already positive and refused to disclose their HIV-positive status during the survey through the unique ID.

Response: Thank you for your comments. Yes, NAIIS checked for blood ARVs but unfortunately the results of the blood ARVs were received after the survey (about 6 months later). On the other hand, LTC and treatment initiation were done in the field almost in real-time during the survey, hence we relied on self-reported ARV use. Nevertheless, there is another paper which provided answer to your question and the paper is submitted to PLOS ONE and is undergoing review. The paper provided in-depth review of implication of self-reported HIV status and ARV use. 

• Typographical errors

- Line 52 (2024-2019) should be corrected. Thank you for your observation. This has been corrected to (2004 – 2019)

- Reference in line 241 (154) should be corrected. Thank you for your observation. This has been corrected in line 307 (now 12 and not 154)

-

---

## [Editor Report · Decision Letter 1]

2 Sep 2021

Optimizing community linkage to care and antiretroviral therapy Initiation: Lessons from the Nigeria HIV/AIDS Indicator and Impact Survey (NAIIS) and their adaptation in Nigeria ART Surge.

PONE-D-21-15583R1

Dear Dr. Jahun,

We’re pleased to inform you that your manuscript has been judged scientifically suitable for publication and will be formally accepted for publication once it meets all outstanding technical requirements.

Kind regards,

Professor Kwasi Torpey, MD PhD MPH

Academic Editor

PLOS ONE
---

## [Editor Report · Acceptance letter]

8 Sep 2021

PONE-D-21-15583R1 

Optimizing community linkage to care and antiretroviral therapy Initiation: Lessons from the Nigeria HIV/AIDS Indicator and Impact Survey (NAIIS) and their adaptation in Nigeria ART Surge. 

Dear Dr. Jahun:

I'm pleased to inform you that your manuscript has been deemed suitable for publication in PLOS ONE. Congratulations! Your manuscript is now with our production department. 

Kind regards, 

on behalf of

Professor Kwasi Torpey 

Academic Editor

PLOS ONE